# Targeted Sequencing of Plasma-Derived vs. Urinary cfDNA from Patients with Triple-Negative Breast Cancer

**DOI:** 10.3390/cancers14174101

**Published:** 2022-08-24

**Authors:** Henrike Herzog, Senol Dogan, Bahriye Aktas, Ivonne Nel

**Affiliations:** 1Department of Gynecology, Medical Center, University of Leipzig, 04103 Leipzig, Germany; 2Soft Matter Physics Division, Peter-Debye-Institute, University of Leipzig, 04103 Leipzig, Germany

**Keywords:** cell-free DNA, liquid biopsy, triple-negative breast cancer, targeted sequencing, somatic variants, plasma, urine

## Abstract

**Simple Summary:**

Circulating cell-free DNA displays vast potential to capture the entire genetic landscape of a tumor and to characterize intratumoral heterogeneity, providing a minimally invasive alternative to tissue biopsy. Several studies have demonstrated the potential of cell-free DNA in the plasma of breast cancer patients. In contrast, very little is known about the utility of urine as an even more patient-convenient analyte for these applications. In this pilot study, we investigated plasma-derived and matching urinary cell-free DNA samples obtained from 15 presurgical triple-negative breast cancer patients using a targeted sequencing approach to identify breast-cancer-related genetic alterations in both body fluids. Taken together, our results indicated that both body fluids appear to be valuable sources bearing complementary information concerning the genetic tumor profile, which might be relevant for disease monitoring and individual treatment decisions.

**Abstract:**

In breast cancer, the genetic profiling of circulating cell-free DNA (cfDNA) from blood plasma was shown to have good potential for clinical use. In contrast, only a few studies were performed investigating urinary cfDNA. In this pilot study, we analyzed plasma-derived and matching urinary cfDNA samples obtained from 15 presurgical triple-negative breast cancer patients. We used a targeted next-generation sequencing approach to identify and compare genetic alterations in both body fluids. The cfDNA concentration was higher in urine compared to plasma, but there was no significant correlation between matched samples. Bioinformatical analysis revealed a total of 3339 somatic breast-cancer-related variants (VAF ≥ 3%), whereof 1222 vs. 2117 variants were found in plasma-derived vs. urinary cfDNA, respectively. Further, 431 shared variants were found in both body fluids. Throughout the cohort, the recovery rate of plasma-derived mutations in matching urinary cfDNA was 47% and even 63% for pathogenic variants only. The most frequently occurring pathogenic and likely pathogenic mutated genes were NF1, CHEK2, KMT2C and PTEN in both body fluids. Notably, a pathogenic CHEK2 (T519M) variant was found in all 30 samples. Taken together, our results indicated that body fluids appear to be valuable sources bearing complementary information regarding the genetic tumor profile.

## 1. Introduction

Breast cancer is a very heterogeneous disease and can be classified into distinct molecular subtypes. The most aggressive subtype is triple-negative breast cancer (TNBC), which is characterized by the lack of expression of estrogen receptor (ER), progesterone receptor (PR) and human epidermal growth factor receptor 2 (HER2). The majority of TNBCs are of high grade and show a high proliferation rate. TNBC accounts for about 15% of breast cancer patients, but approximately 25% of breast-cancer-related deaths [1,2].

Over the last few years, significant progress has been made in the field of targeted therapies for some breast cancer subtypes, such as endocrine therapy for hormone-receptor-positive tumors or trastuzumab for tumors with HER2 overexpression. In contrast, patients with TNBC cannot benefit from these advancements as they lack the biological targets. Hence, chemotherapy, in some occasions combined with Pembrolizumab, a monoclonal antibody against anti–programmed death 1 (PD-1), remains the leading therapy option for TNBC [3,4]. However, only one-third of the patients achieve pathological complete response (pCR) after neoadjuvant chemotherapy (NACT) and patients have an increased risk of recurrence and poor prognosis [5,6].

Currently, pCR is the most important prognostic factor in TNBC and those patients can benefit from an improved outcome if histopathological staining after NACT does not display tumor cells [7,8,9]. Particularly, patients that do not reach pCR after chemotherapy might require targeted treatment approaches.

The genetic profiling of TNBC revealed potential options for tailored therapy strategies, such as modified chemotherapy approaches targeting the DNA damage response, angiogenesis inhibitors, immune checkpoint inhibitors or even anti-androgens, which are currently under investigation in various phase I to III clinical studies [10,11,12,13]. TNBC patients and patients with a hereditary disease (about 20% of the patients) receive multi-germline gene panel testing for risk assessment. Subsequently, therapeutic decisions are based on mutation profiles that were received from an initial tissue biopsy or blood sample. During therapy, however, the genetic tumor profile might change, meaning that distinct mutations such as targetable driver mutations might become functionally neutral passenger mutations [14]. Consequently, treatment efficacy might be affected, ultimately leading to resistance.

Based on histology, the majority of patients are diagnosed with early-stage disease without macroscopic evidence of metastases [15]. However, several studies have shown that residual micrometastases exist even after initial treatment and may result in disease relapse [16,17]. Cell-free circulating DNA (cfDNA) became a promising analyte in oncology research. Particularly, DNA fragments originating from the tumor cells, so-called circulating tumor DNA (ctDNA), display vast potential for real-time monitoring of the tumor, as they mirror tumor heterogeneity, including genetic information from all subclones and possible micrometastases that are not represented by a single tissue biopsy [18]. As ctDNA is postulated to be shed by all tumor sites, even micrometastases, it could enable the early detection of recurrence as well as the identification of somatic molecular alterations during treatment courses that might be relevant for targeted therapy options [19].

One promising approach to identify tumor-specific alterations in ctDNA might be the targeted sequencing of cancer-related mutations in the primary tumor tissue and verification of them in ctDNA at baseline. Common and reliable methods for mutational analysis are digital droplet PCR (ddPCR) or the more comprehensive next-generation sequencing (NGS). Many studies compared mutational profiles in plasma-derived ctDNA at baseline with matched tissue samples from breast cancer patients and found good correlations [20].

In contrast to plasma-derived ctDNA, only a few studies were performed using urinary ctDNA from patients with breast cancer, and they revealed that targeted NGS appeared to be a sensitive method to detect tumor-specific genetic features [21]. Here, ctDNA derived from patients’ urine might offer promising opportunities for non-invasive monitoring at frequent intervals with low effort. However, very little is known about the utility of urinary cfDNA for genetic sequencing to find targetable mutations and gain access to tailored therapeutic strategies. In this pilot study, we investigated plasma-derived and matching urinary cfDNA samples obtained from 15 presurgical TNBC patients using a targeted sequencing approach to identify breast-cancer-related genetic alterations in both body fluids. We aimed to examine the concordance between genetic alterations found in both body fluids to assess whether they might provide similar or complementary information regarding the genetic profile of the tumor. Further, this study was thought to investigate the potential of urinary cfDNA for clinical usage.

## 2. Materials and Methods

### 2.1. Patient Characteristics

The study was conducted at the Department of Gynecology at the University Hospital Leipzig, Germany. After agreeing and signing a written informed consent in accordance with the requirements of our institution’s board of ethics (internal reference number: No. 216/18-ek), blood and urine from patients with histopathologically confirmed TNBC were sampled prior to surgery. Clinico-pathological patient data were collected from medical records and patient characteristics are listed in Table 1. In total, 15 female TNBC patients were enrolled in the study. At diagnosis, core needle biopsies were obtained. The median Ki-67 proliferation rate was 75% throughout the cohort (ranging from 10–90%). All patients were negative for estrogen receptor (ER), progesterone receptor (PR) and human epidermal growth factor receptor 2 (HER2). Among the TNBC cohort, the median age at sample withdrawal was 48 years, ranging from 26 to 69 years. Almost all patients (*n* = 14; 93.3%) presented with early-stage BC and the majority (*n* = 13; 86.6%) received neoadjuvant chemotherapy (NACT) to shrink the size of the primary tumor before resection. Local lymph node involvement was found in 3 of 15 patients (20%) and none of the patients had evidence of distant metastatic disease. The majority of the patients (*n* = 11; 73.3%) presented with grade 3 tumors. Both plasma and urine samples were obtained from each patient, resulting in a total number of 30 liquid biopsy samples that were subjected to cfDNA analysis and subsequent identification of somatic mutations.

For risk assessment and possible enrollment in intensified after-care programs, the TNBC patients received a germline gene panel testing using the TruRisk test including ATM, BARD1, BRCA1, BRCA2, BRIP1, CDH1, CHEK2, PALB2, RAD51C, RAD51D, TP53 and PTEN. More than half of the patients (*n* = 9; 60%) revealed known germline mutations in the BRCA1/2 genes. One patient presented an additional PALB2 germline mutation and another patient had a germline variant of uncertain significance in the BRIP1 gene. Germline testing was missing in only one of the cases.

### 2.2. Acquisition and Processing of Blood and Urine Samples

For this pilot study, blood and urine samples were obtained one day prior to primary tumor resection. Matching blood and urine specimens were collected and processed at the same time to assure comparability. From each patient, 9 mL of EDTA blood (S-Monovettes^®^, Sarstedt, Germany) and 10 mL of urine (Urine-Monovettes^®^, Sarstedt, Germany) were collected and processed within 4 hrs. to avoid DNA degradation. Blood samples were centrifuged at 4000× *g* for 10 min to separate plasma from peripheral blood cells. Urine samples were centrifuged at 4000× *g* for 10 min to remove any cellular debris. Plasma and urine supernatants were stored at −80 °C until cfDNA extraction.

### 2.3. Isolation and Quantification of Cell-Free DNA (cfDNA)

Aliquots of plasma and urine samples were thawed immediately and only once prior to cfDNA extraction to minimize freeze–thaw effects [22]. Isolation and quantification of cfDNA was performed as described elsewhere [23]. Subsequently, cfDNA was isolated from 4 mL of plasma and 10 mL of matching urine supernatant for each patient using the QIAamp MinElute ccfDNA Midi Kit (QIAGEN, Hilden, Germany) according to the manufacturers’ instructions. To obtain a maximum yield of nucleic acids, cfDNA from plasma was eluted in 50 µL and cfDNA from urine in 20 µL ultraclean water, respectively. Concentrations of the isolated cfDNA were quantified fluorometrically using a Qubit 2.0 fluorometer and the Qubit dsDNA HS Assay kit (Cat. No. Q32851, Invitrogen; Thermo Fisher Scientific, Inc., Waltham, MA, USA), following the manufacturers’ instructions, and samples were stored at −20 °C prior to targeted sequencing.

### 2.4. Library Preparation and Targeted Sequencing of Breast-Cancer-Related Genes

The library preparation was performed using the QIAseq Targeted DNA Panel Library Prep Kit (QIAGEN, Hilden, Germany). An amount of maximal starting material was enzymatically fragmented. Ends were repaired and 3′ adenylated. Barcoded adapters, including the unique molecular identifier (UMI) and sample-specific indices, were ligated to the overhang and the reactions were cleaned up. Target enrichment was conducted using the QIAseq Human Breast cancer Panel (DHS-001Z, QIAGEN, Hilden, Germany), amplifying all coding regions of 93 genes which were known to be associated with breast cancer. Target regions were enriched by target-specific PCR (6 cycles). After bead-based reaction clean up, molecules were enriched by 22 cycles of universal PCR followed by a final clean-up step. Library preparation was quality controlled using capillary electrophoresis (Agilent DNA 7500 Chip). High-quality libraries were pooled in equimolar concentrations based on the Bioanalyzer automated electrophoresis system (Agilent Technologies, Santa Clara, CA, USA). The library pools were quantified using qPCR and entirely subjected to cluster generation. Pooled libraries were analyzed by paired-end sequencing on an Illumina NextSeq550 instrument (Illumina Inc., San Diego, CA, USA), using a NextSeqv2.5 High Output 300 bp cassette (2 × 150 bp, 2 × 8 bp), according to the manufacturers’ instructions. The above-described library preparation and targeted sequencing procedure was performed on a fee-for-service basis in the Genomic Services laboratories by QIAGEN, Hilden, Germany.

### 2.5. Sequencing Parameters

After cfDNA extraction, samples were subjected to targeted sequencing of breast-cancer-related genes to identify ctDNA and genetic alterations thereof. The concentration of ctDNA in the enriched libraries ranged from 10.22–71.3 nM (median: 52.22 nM). A median of 18,081,706 total read fragments per sample was obtained (range: 7,091,278−38,001,200). The median targeted sequencing coverage was 201-fold for plasma (range: 43.79–462.18) and 67-fold for urine samples (range: 6.67–1128.36). A median of 99.29% of all reads was successfully mapped to the reference genome hg19 (range: 97.72−99.43%). A table listing comprehensive sequencing parameters can be found in the Appendix A (Appendix A).

### 2.6. Bioinformatical Analysis and Filtering of Somatic Variants in ctDNA

The CLC Genomics Workbench and the Biomedical Genomics Analysis plugin (QIAGEN, Hilden, Germany) were used for initial data analysis, such as an assessment of library yield, the number of total reads and percentage of mapped reads followed by UMI analysis, quality control and base calling. All reads were aligned to the human reference genome hg19/GRCH37 to identify genetic alterations in comparison to the human reference genome. The integration of UMIs to each fragment assured that only true positive variants were identified and helped to reduce library bias as well as sequencing artifacts [24]. The rate of false-positive variants was reduced while increasing the sensitivity of variant detection at allele frequencies down to 0.1%. Having in mind that a low read amount might occur due to bad library quality, we ensured that all samples had more than 5 million read fragments [25].

Subsequently, the QCI Interpret Translational program (QIAGEN, Hilden, Germany) was applied for annotation, filtering, classification and interpretation of the called variants. Application of the tool enabled us to evaluate the detected genomic variants in the context of professional association guidelines, annotations, various publicly accessible databases, drug label data, published biomedical evidence and clinical trials.

Unlike other variant annotation and scoring tools that tend to exploit a single information type and are restricted in scope, QCI uses a Combined Annotation Dependent Depletion (CADD) framework that integrates multiple annotations into one metric. In silico algorithms that perform a prediction of the effect that a variant has on the amino acid and the resulting protein are presented for each detected variant [26]. Computational evidence makes predictions about whether a missense change is located in the conservative region of the protein, whether it is damaging to the resulting protein in terms of structure and function, and whether there is an effect on splicing. There are numerous algorithms for missense prediction, such as PolyPhen-2, SIFT, MutationTaster or MutationAssessor [27,28,29,30]. Further, based on the ACMG/AMP guidelines (2015) [31], all variants were classified into five groups, according to their impact on the biological function of a protein: pathogenic, likely pathogenic, uncertain significance, likely benign and benign. Additionally, variants were assigned to four tiers regarding their clinical significance as described in the AMP/ASCO/CAP guidelines (2017) [32].

We used a sequential filter cascade to characterize the detected variants.

First, the confidence filter was set up to exclude variants of a low quality. All variants with a call quality below 50 were filtered out, indicating very high confidence that the remaining variants are truly existing in the sample rather than resulting from sequencing errors such as amplification errors. Additionally, variants with an allele frequency (VAF) of less than 3% were filtered out to further eliminate variant positions with low levels of support [33,34]. Only variants not occurring in the top 5% of the most exonically variable 100-base windows among healthy public genomes remained in the list.

Secondly, the common variants filter excluded all variants with a prevalence of >1% in the normal population using reference databases such as Allele Frequency Community (gnomAD&CGI), ExAC, NHLBI ESP exomes and the 1000 Genomes Project, unless it was an established pathogenic variant, to exclude variants that are deemed polymorphic or benign and thus are most likely not disease-associated. The cut-off value was chosen according to the AMP/ASCO/CAP guidelines 2017 [32], which recommended a primary cut-off at 1% in absence of normal paired tissue.

Thirdly, the predicted deleterious filter only kept variants that were located no more than 20 bases into the intron. Furthermore, the filter only retained variants that were additionally classified to be pathogenic, likely pathogenic or of uncertain significance (according to the ACMG guidelines classification), or that were gain-of-function-associated, or variants that were associated with loss of function of a gene, including frameshift, in-frame indel, start/stop codon changes, missense, copy number loss, splice site loss up to 2 bases into the intron, or as predicted by MaxEntScan (G. Yeo and C.B. Burge, available at: http://hollywood.mit.edu/burgelab/maxent/Xmaxentscan_scoreseq.html (accessed on 21 August 2022)), a scoring tool for human splice sites using the Maximum Entropy Principle.

The original raw sequencing data that support the findings of this study have been uploaded as fastq files (forward and reverse) and are available at the Sequence Read Archive (SRA) at GeneBank under the BioProject ID PRJNA844219.

### 2.7. Statistical Analysis

Categorical data are reported as percentages. For quantitative data, median and range are presented. The Spearman’s rank correlation coefficient was calculated to evaluate the association between plasma and urine cfDNA concentration. The difference between plasma and urine cfDNA concentration was analyzed by the Wilcoxon signed-rank test. The Chi-square test was used to test the association between categorical data. *p*-values less than 0.05 were considered statistically significant. All statistical analyses were performed using IBM SPSS Statistics (version 28.0.0.0, IBM, Armonk, NY, USA) and R code Jamovi (The jamovi project (2021), jamovi (version 2.0), Retrieved from https://www.jamovi.org).

## 3. Results

### 3.1. Cell-Free DNA (cfDNA) Concentration in Matched Plasma and Urine Samples

The median cfDNA concentration from 4 mL plasma eluted in 50 µL ultraclean water was 172 ng/mL, ranging from 37–442 ng/mL. The median cfDNA concentration from 10 mL urine eluted in 20 µL was 196 ng/mL, ranging from 0–1730 ng/mL. In two cases, no urinary cfDNA was detectable using the Qbit fluorometer, but a sufficient concentration was obtained after target amplification. However, coverage remained low for these two samples. Some cases revealed remarkable differences between plasma-derived and urinary cfDNA concentrations. In addition, high inter-individual variability was observed (Table 2). Thus, no significant association was revealed between cfDNA concentrations in plasma and the matching urine samples (Spearman’s correlation coefficient r = −0.187, *p* = 0.504). Looking at the difference between plasma and urine cfDNA concentrations, the Wilcoxon signed-rank test showed no significant difference (median 172 ng/mL vs. 196 ng/mL; *p* = 0.117).

### 3.2. Investigation of Somatic Variants in ctDNA Derived from Matched Plasma and Urine Samples

After the application of our personalized filter cascade, a total number of 3339 somatic breast-cancer-related genetic alterations (VAF ≥ 3%) were detected in the cohort, whereof 1222 vs. 2117 variants were found in plasma-derived vs. urinary cfDNA, respectively. The detection of breast-cancer-associated gene alterations confirmed that the analyzed cell-free DNA fragments most likely originated from the tumor. Therefore, from now on we will refer to it as circulating tumor DNA (ctDNA). Of all somatic variants, 791 vs. 1686 were exclusively found in plasma-derived respective urinary ctDNA when comparing both cfDNA sources per patient. Further, a total number of 431 shared variants were found in ctDNA derived from both body fluids (Figure 1). A table listing the number of exclusive and shared variants per patient can be found in the Appendix A (Appendix A). Per sample, we found a median number of 66 variants in plasma vs. 110 variants in urinary ctDNA. The number of variants detected per plasma sample ranged from 46 to 261 and from 36 to 320 per urine sample (Figure 2). Interestingly, a median number of 31 shared variants (range: 4–46) were found per matched ctDNA sample pair. Notably, the number of variants found in urinary ctDNA was significantly higher compared to the number of variants comprised in plasma-derived ctDNA (median 110 vs. 66; *p* = 0.048).

### 3.3. Abundance of Detected Variants in Plasma and Urinary ctDNA

At the time being, it is common to use the variant allele frequency (VAF) as a quantitative measure of a tumor-specific variant. It is expressed as the ratio between the number of mutated and wild-type DNA copies [35]. Hence, VAF is the percentage of sequence reads observed, matching a specific DNA variant divided by the overall coverage at that locus [36]. If reference material is not available, VAF might be used to distinguish somatic and germline mutations. The latter ones are present with either 50% if heterozygous or 100% if homozygous [37]. However, somatic variants which are acquired usually present lower VAF, because they do not occur in all cells. For many tumor sequencing applications, 5% VAF mutation sensitivity is sufficient [38]. Here, we separated all variants into five groups: VAF 3 ≤ 5%, 5 ≤ 10%, 10 ≤ 20%, 20 ≤ 50% and 50–100%. The distribution of allele frequencies separated for plasma and urine is shown in Figure 3. Of all variants isolated from the ctDNA of both body fluids, 1226 (37%) had a very low abundance with VAF of 3 ≤ 5%, and 1061 (32%) variants showed a low abundance with a VAF in the range of 5 ≤ 10%. Further, 633 (19%) variants appeared with an intermediate abundance which means a VAF of 10 ≤ 20%. In addition, 346 (10%) respective. 73 (2%) variants were of high and very high abundance with a VAF of 20 ≤ 50% and 50–100%, respectively. Even though we did not have matched germline samples for our cohort, 98% of the variants (*n* = 3266) displayed a VAF below 50%. Of all variants, 88% (*n* = 2920) presented a low abundance (VAF below 20%), indicating that the variants are rather of somatic than germline origin.

### 3.4. Classification of Somatic Variants According to Biological Impact

As mentioned in the methodology section, all somatic variants were classified into five groups, according to their impact on the biological function of a protein encoded by the mutated gene. Usually, a variant’s impact on the amino acid changes is categorized as deleterious, medium, benign or neutral. Depending on the changes, a mutation might cause in the structure and therefore the function of the protein expressed by the gene the variant is located at; variants can be divided into various impact groups: pathogenic, likely pathogenic, variant of uncertain significance (VUS), likely benign and benign. In our TNBC cohort, the distribution of the variants into the five different groups was very similar for somatic variants derived from plasma compared to urinary ctDNA (Figure 4). In both body fluids, most variants were classified as VUS. A total number of 1022 (83.6%) variants from plasma-derived ctDNA and 1819 (85.9%) variants from urinary-derived ctDNA were categorized into this group. The second most frequently represented group was the likely pathogenic category, comprising 98 (8.0%) variants in plasma ctDNA and 204 (9.6%) variants in urinary ctDNA. Further, 77 (6.3%) of all variants found in plasma and 71 (3.4%) of all urinary ctDNA variants were categorized as pathogenic. Since variants with a prevalence of >1% among the normal population were filtered out using the common variant filter, the number of remaining likely benign and benign variants was supposed to be low. Hence, of all plasma-derived variants, only 20 (1.6%) variants were classified as likely benign and 5 (0.4%) variants belonged to the benign group. Looking at the ctDNA variants found in urine, numbers were equally low for these two categories: 17 (0.8%) likely benign and 6 (0.3%) benign variants. When looking at the pie charts, the distribution of the somatic variants into the five categories appears to be similar for both body fluids (Figure 4). However, the distribution is significantly different for plasma and urinary ctDNA (*p* < 0.001), most conspicuous in the pathogenic category (6.3% vs. 3.4%).

Per plasma sample, median numbers of 5 pathogenic, 6 likely pathogenic, 53 VUS, 1 likely benign and 1 benign variant were found in the ctDNA. Having a look at the variants detected in urinary ctDNA, median numbers of 5 pathogenic, 10 likely pathogenic, 96 VUS, 1 likely benign and 1 benign variant were detected.

Interestingly, when comparing all variants found in plasma vs. urinary ctDNA for each patient, a median of 46.8% of (range: 4.9–86.8%) plasma ctDNA variants were recovered in the matching urine ctDNA samples. Vice versa, only a 15.7% recovery rate (range: 5.9–86.1%) of urinary ctDNA variants in plasma was observed throughout the cohort. Remarkably, when only looking at the pathogenic variants, an even higher recovery rate could be found. Here, a median of 62.5% of plasma-derived ctDNA variants could be found in the matching urine samples. The other way round, a median of 66.7% of pathogenic variants found in urinary ctDNA were also present in the paired plasma samples.

### 3.5. Classification of Somatic Variants According to Clinical Significance

As mentioned above, variants were additionally assigned to four tiers regarding their evidence-based clinical significance in cancer diagnosis, prognosis and/or therapeutics. Variants in tier I are of strong clinical significance (level A and B evidence), variants in tier II are of potential clinical significance (level C and D evidence), variants in tier III are of unknown clinical significance and variants in tier IV are characterized to be benign or likely benign [32]. In our TNBC cohort, 1059 (86.7%) of the somatic variants found in plasma-derived ctDNA were classified as tier III. Further, 130 (10.6%) variants were of potential clinical significance, meaning they belong to tier II and 27 variants (2.2%) were described to be benign or likely benign (tier IV). Only 6 variants (0.5%) were categorized into tier I, meaning they are of strong clinical significance. Having a look at the classification of urinary ctDNA variants, the distribution into the four tiers was similar. Here, 1924 (90.9%) of the variants were assigned to tier III, while 158 variants (7.5%) were classified as tier II and 24 variants (1.1%) as tier IV. Only 11 variants (0.5%) were described to be of strong clinical significance (tier I; Figure 5). Variants belonging to tier I were found in only 13 of all 30 samples and were solely BRCA1/BRCA2 alterations. Notably, the percentage of variants belonging to tier III (uncertain clinical significance) is very similar to the number of variants classified as VUS in the biological impact-based classification described above.

### 3.6. Analysis of Most Frequently Mutated Breast-Cancer-Related Genes in Plasma vs. Urinary ctDNA

As mentioned earlier, ctDNA obtained from plasma and urine was investigated for the presence of variants in 93 breast-cancer-related genes. Throughout our cohort of TNBC patients, almost 20% (*n* = 637) of all detected somatic variants in both body fluids were located in the MUC16 gene. Moreover, 259 variants were found in the KMT2C gene followed by 194 variants in the NF1 gene. Interestingly, MUC16, KMT2C, NCOR1 and CHEK2 were altered in all 30 liquid biopsy samples. The ten most frequently altered genes in both ctDNA sources are shown in Figure 6.

The distribution of the top 10 mutated genes among all of the above-mentioned impact categories varied between both ctDNA sources. The number of variants is affected by parameters such as protein size. For example, MUC16 is a giant protein consisting of around 22,000 amino acids, which poses a high risk of residue alterations because of random DNA repair errors [39]. However, when focusing on pathogenic and likely pathogenic variants, the picture became clearer. We exclusively analyzed pathogenic and likely pathogenic variants among the 15 TNBC patients and found that 59 different genes displayed at least one (likely) pathogenic variant. Notably, each patient presented with at least one pathogenic or likely pathogenic variant. Among the cohort, NF1 was by far the most frequently altered gene regarding these two categories, which resulted in 49 vs. 54 (likely) pathogenic variants in plasma- vs. urine-derived ctDNA, respectively, followed by KMT2C (25× vs. 32×), CHEK2 (25× vs. 9×) and PTEN (25× vs. 13×) (Table 3).

### 3.7. Occurrence of Pathogenic and Likely Pathogenic Variants among the Most Frequently Altered Genes in the TNBC Cohort

As ctDNA can be altered at multiple locations within a gene, we were interested in the specific variants that were frequently occurring in our cohort. Here, we focused on the pathogenic/likely pathogenic variants, as these mutations are known to have an effect on the protein function and might therefore be disease-associated. Interestingly, our analysis revealed not only commonly mutated genes associated with breast cancer as defined by the sequencing panel, but also some specific variants that occurred very frequently in our TNBC cohort. Notably, a pathogenic CHEK2 (T519M) variant was found in all 30 samples. Two pathogenic NF1 variants, T467I and c.2325+3A>G, were comprised in 13 (86.7%) vs. 15 (100%) of plasma-derived ctDNA samples and in 9 (60.0%) vs. 10 (66.7%) of urinary-derived ctDNA samples, respectively. A pathogenic PTEN variant (C136Y) was detected in 12 (80.0%) vs. 7 (46.7%) of all plasma vs. urine samples. Additionally, there were three common likely pathogenic variants throughout the cohort. A KMT2C variant (W1056*) and a NF1 variant (L792F) were found in ctDNA of all 15 plasma samples as well as in 13 rsp. 9 of the urine samples. Further, a likely pathogenic PTEN variant (V133I) was present in 12 (80%) of the plasma-derived and in 6 (40%) of the urinary-derived ctDNA samples. A heatmap displaying the occurrence of each pathogenic variant throughout the plasma and urine samples of our TNBC cohort is shown in Figure 7. In addition, we found frequently occurring variants that were classified as VUS in our cohort. For example, a MUC16 variant (V13007G) was detected in all 15 plasma samples and in 14 of 15 urine samples. Further, a NF1 (F384L) and a CHEK2 (E493K) variant were found in all plasma and 11 urine samples.

## 4. Discussion

This pilot study was thought to investigate the potential of urinary cfDNA for clinical usage. We investigated plasma-derived and matching urinary cfDNA samples obtained from 15 presurgical TNBC patients using targeted sequencing to identify common breast-cancer-related genetic alterations in both body fluids. In contrast to plasma-derived cfDNA, only a few studies were performed using urinary cfDNA from patients with breast cancer. Focusing on the PIK3CA mutation, Liu et al. investigated correlations between cfDNA derived from plasma vs. urine of 200 patients with early breast cancer [40]. They reported a lower mean concentration of cfDNA extracted from urine compared to plasma with a strong correlation. In our cohort, urine samples resulted in slightly higher median cfDNA concentration with increased variability compared to plasma samples without statistically significance though. The low correlation coefficient between cfDNA concentrations in plasma and the matching urine samples indicated that low plasma cfDNA levels were not necessarily associated with low levels of urinary cfDNA, enhancing the complementary nature of both body fluids. However, it has to be considered that the sample size was very low. In their study, Liu et al. also compared urinary cfDNA of breast cancer patients with healthy volunteers and found significantly elevated cfDNA levels in the patient group. In a study by Zuo et al, the mean concentration of cfDNA detected in 50 healthy controls was reported to be as high as 45.2 ng/mL (standard deviation: 7.1 ng/mL). Further, in 250 breast cancer patients, the mean concentrations of plasma-derived vs. urinary cfDNA were 1.72-fold vs. 2.73-fold higher compared to the controls (*p* < 0.001) [41]. In accordance with these findings, preliminary experiments in our laboratory with urine samples of 20 healthy female individuals showed cfDNA levels in the range of 19.87–91.43 ng/mL (mean: 49.42 ng/mL; data not shown). In our study, the median urinary cfDNA concentration was higher compared to plasma-derived cfDNA levels (196 vs. 172 ng/mL). It is already well known that the concentration of cfDNA in blood varies significantly between individuals and ranges between 0–5 and >1000 ng/mL in patients with cancer and between 0 and 100 ng/mL in healthy subjects [18]. Usually, cfDNA yields are higher in patients with malignant lesions compared to patients without tumors. Nonetheless, elevated levels of cfDNA might also reflect physiological and non-malignant pathological events including diabetes, inflammation, tissue trauma or sepsis [42,43]. Further, the amount of cfDNA might be influenced by other factors that vary greatly between individuals, such as clearance, degradation and other filtering events of the blood and lymphatic circulation [44]. Several studies have shown a correlation between cfDNA levels and the outcome as well as survival of cancer patients. In addition, ctDNA levels have been correlated with the size and stage of the tumor, which implied that the cfDNA concentration might also be affected by tumor burden [18]. However, cfDNA is also released by normal cells such as stromal or hematopoietic cells, hence the ctDNA only represents a part of the entire cfDNA concentration. As it depends on many individual factors, the cfDNA concentration alone should not be over-prioritized. Changes in cfDNA concentration should be rather evaluated in the course of therapy and in combination with other cfDNA characteristics such as the mutational profile.

Recently, a study by Guan et al. described the utility of urinary ctDNA to monitor recurrence in 300 patients with early breast cancer using serial sampling before treatment and at various time points during follow up [45]. They found detectable mutations in 38% of the patients and the agreement with matched tissue samples was 97%, which was congruent with results by Zuo et al., who reported a 97% agreement of PIK3CA mutation profiling in plasma and urinary ctDNA compared with respective tissue samples [41].

In our study, primary tissue was not analyzed, as the tumor profile might change during therapy and we focused on the utility of liquid biopsies. The majority of our cohort received NACT; hence, the ctDNA collected upfront primary tumor resection might reflect a real-time picture of the genetic profile, which could be useful to detect recurrence at a molecular level before clinical symptoms occur. In cancer patients, three cellular sources of cfDNA can be found: normal extratumoral cells, malignant cells and tumor microenvironmental cells. Each of these compartments may be exposed to various biological processes that release different forms of DNA into the circulatory system [46]. Thus, a significant part of total cfDNA consists of non-mutated DNA, but according to various studies, the fraction of ctDNA accounts for ~0.1–89% of cfDNA [18]. As the detection of breast-cancer-associated gene alterations confirmed that the analyzed cfDNA fragments most likely originated from the tumor, it is appropriate to assume that the total mutant cfDNA fraction accounts for the ctDNA derived from malignant cells from the tumor or possible micrometastases [47]. Notably, per patient we detected a median number of 66 variants in plasma vs. 110 variants in urinary ctDNA. We revealed that 48 of all 431 shared mutations found in both body fluids were pathogenic, resulting in a median number of 4 shared pathogenic variants per sample. The recovery rate of pathogenic plasma-derived variants in urinary ctDNA was 63%, and vice versa, 66% of pathogenic urinary variants could be detected in the matched plasma samples. In other words, our results showed that about two-thirds of the pathogenic variants from plasma-derived ctDNA could be found in urinary ctDNA as well. Further, 33 of the shared mutations were likely pathogenic. Taking all somatic variants into consideration, 47% of plasma-derived variants were found in the matching urinary ctDNA and only 16% of urinary variants occurred in plasma-derived ctDNA. This finding indicated the complementary value of both liquid biopsy sources as not only pathogenic variants are of interest. When classifying the detected somatic variants according to their biological impact, we found 6 vs. 3% pathogenic variants in matching plasma and urinary ctDNA samples and 8 vs. 10% likely pathogenic variants and the majority, 84 vs. 86%, were of uncertain significance, hence bearing great potential for the discovery of novel targets by further investigation of these variants. Only 2% vs. 1% were likely benign and benign variants, confirming the soundness of the applied filtering cascade. Fittingly, subsequent analysis of the clinical significance revealed that 87 vs. 91% of the detected somatic variants in plasma vs. urinary ctDNA were of unknown clinical significance and thus belonged to tier III. This is reasonable because unknown variants cannot be under investigation in clinical trials yet, thus lacking any clinical evidence. Further, 11 vs. 7% were of potential clinical significance (tier II) and most likely resulting from the variants classified as pathogenic and likely pathogenic variants according to their biological impact. Only 2% vs. 1% of the variants belonged to tier IV, probably from the few remaining likely benign and benign variants. In fact, 6 vs. 11 variants, detected in 13 of 30 samples, were of strong clinical significance (tier I) in plasma vs. urinary ctDNA. On the one hand, it makes sense that they were solely BRCA1 and BRCA2 alterations. DNA repair mechanisms are playing a crucial role regarding sensitivity against platinum-based chemotherapy such as NACT. Double-strand breaks are mainly repaired by homologous recombination, which is regulated by BRCA1 and BRCA2 expression. Among TNBC, a variety of subgroups have been identified, including deficiency in homologous recombination, which was partly associated with a loss of BRCA1 or BRCA2 function [10,48]. Further, one major repair mechanism of single-strand breaks is the base excision pathway in which Poly (ADP-ribose) polymerase (PARP) plays a key role. It cannot be excluded that germline and somatic BRCA1/2 alterations might be biologically equivalent [49]. It was reported in the literature that the genetic profile of a TNBC sub-cohort without a germline BRCA1/2 mutation was similar to those with a germline BRCA mutation [50]. In a study by Davies et al., a test (HRDetect) was described that might predict BRCA1/2 deficiency based on mutational signatures [51]. They identified 12.4% of breast cancers as being BRCA1/2-deficient although they did not present a BRCA1/2 germline mutation. Due to various clinical trials of PARP inhibitors in metastatic TNBC and germline as well as somatic BRCA1/2 alterations (e.g., NCT02401347 and NCT03330847), these variants are well-described and of strong clinical significance. On the other hand, these tier I-classified BRCA1/2 alterations might be germline mutations missed by the filter steps applied during variant calling. However, this is rather unlikely as 60% of the patients had BRCA1/2 germline mutations in the initial blood sample drawn at diagnosis. In case of a filtering failure, we would expect at least 18 positive samples.

Our study was limited not only by small sample size, but also by a strongly varying sequencing coverage between patients, and even more importantly, between the plasma and urine samples of individual patients, which were compared. The coverage was 201-fold for plasma and 67-fold for urinary ctDNA. It was controversially discussed in the literature which coverage is sufficient for translation into the clinic. For targeted sequencing, a high coverage of 1085× was recommended for 2% VAF to reach a sensitivity of 95% and a >95% positive predictive value [52]. The VAF information is important for assessing the distribution of a variant in a patient cohort. Here, the sequencing coverage plays a crucial role as a low coverage might lead to an underestimation of the mutant molecules or an overestimated VAF [35]. Even with adequate coverage, it remains a delicate matter to reliably determine whether a called somatic mutation is a driver or a passenger mutation.

When analyzing the somatic mutations, we decided to focus on the classification of variants according to their impact on the biological function of a protein to ensure comparability with other studies, as most publications contain these terms. At the time being, hardly any driver mutations are known for TNBC [53,54]. Hence, no actionable mutations with approved targeted treatment options were available for patients in this cohort. Therefore, a classification according to the clinical significance of the variants appeared to be inappropriate.

Comparing both ctDNA sources, we found variants of all impact categories among the respective top 10 altered genes. However, when focusing on the (likely) pathogenic variants, we discovered four genes, which were most frequently altered in plasma-derived and urinary ctDNA. Among our cohort, the most frequently occurring pathogenic and likely pathogenic mutated genes were NF1, CHEK2, KMT2C and PTEN in both body fluids. Notably, a pathogenic CHEK2 variant (T519M) was found in all 30 samples and two pathogenic NF1 variants (T467I + c.2325+3A>G), as well as a pathogenic PTEN variant (C136Y), were also very frequent in the cohort.

NF1 mutations have been reported in 2.2% (7/313) of TNBC samples analyzed in the Catalogue of Somatic Mutations in Cancer (COSMIC, May 2020). NF1 encodes Neurofibromin, a GTPase-activating protein that acts as a tumor suppressor by depressing Ras signaling. Hence, loss of neurofibromin function may result in increased signaling through the Ras pathway and downstream MAPK and mTOR pathways. Tumors bearing NF1-inactivating mutations may therefore be sensitive to mTOR inhibitors and MAPK pathway inhibitors, such as MEK inhibitors [55]. The pathogenic alteration reported here (c.2325+3A>G) occurs near the splice junction between transcribed exons 19 and 20. Splice site mutations may lead to exon skipping or protein truncation. Therefore, a splicing error at this position may disrupt or remove the majority of the protein, including the GAP-related domain, which is primarily responsible for inhibition of the Ras pathway. NF1 T467I is a missense alteration prior to the GAP-related domain of the neurofibromin protein (InterPro). This alteration has been reported as well (COSMIC, July 2021), but it has not been functionally characterized. Currently, for the pathogenic variants NF1 described above, which were detected among the cohort, there is no approved targeted therapy for TNBC.

PTEN loss or inactivating mutation may lead to increased activation of the PI3K/Akt/mTOR pathway and therefore may lead to uncontrolled cell growth and suppression of apoptosis. Hence, inhibitors of this pathway may be relevant in a tumor with loss or mutation of PTEN [56,57]. The mTOR inhibitors Everolimus and Temsirolimus have been approved by the FDA, EMA and PMDA for use in some indications, and clinical trials of these and other mTOR inhibitors as well as inhibitors of PI3K and Akt are currently underway in multiple tumor types [58,59]. In addition, preclinical studies have shown that PTEN-deficient tumors may be sensitive to PARP inhibitors. Loss of PTEN expression or PTEN mutations were associated with the TNBC subtype more frequently than with HER2- or ER/PR-positive subtypes and have been correlated with breast cancer progression, brain relapse and metastasis [60]. PTEN C136Y is a missense alteration within the phosphatase tensin-type domain of the PTEN protein. This alteration has been reported to lead to a loss of PTEN phosphatase activity.

Further, CHEK2 encodes the tumor suppressor, checkpoint kinase 2 (Chk2), a serine/threonine kinase that plays an important role in cell cycle arrest in response to DNA damage [61,62]. There are currently no approved therapies targeting inactivating alterations in CHEK2. However, depletion of CHEK2 has been reported to increase sensitivity to PARP inhibitors in preclinical models and PARP inhibitors are in clinical trials in cancers with DNA repair deficiencies, including CHEK2 alterations [63] (NCT03786796, 2019; NCT03012321, 2019; NCT02576444, 2019; NCT02401347). CHEK2 mutations have been reported in less than 1% (1/303) of the TNBC samples analyzed in COSMIC (May 2020). For CHEK2 (T476M), no therapies have been approved for TNBC yet. In other indications, Olaparib, Rucaparib, Niraparib and Talazoparib are approved based on somatic alterations. In breast cancer, only Olaparib and Talazoparib are approved for patients harboring germline BRCA mutations [64].

KMT2C, more commonly known as MLL3 (mixed-lineage leukemia 3), encodes a histone methyltransferase enzyme involved in the regulation of gene transcription [65]. There are no therapies specifically targeting MLL3-deficient tumors. Preclinical data suggested that inactivation of another member of the MLL family, MLL, may predict sensitivity to Histone Deacetylase (HDAC) inhibitors, although it is unclear whether this approach would be relevant for MLL3 aberrations [66]. In addition, a preclinical study in acute myeloid leukemia (AML) with MLL3 suppression reported inhibition of cell and tumor growth with a bromodomain and extra-terminal (BET) inhibitor, suggesting another possible therapeutic approach for MLL3 alterations [67]. However, further research is needed to clarify the relevance of this therapeutic approach. KMT2C mutations have been reported in 3.6% (4/111) of the TNBC samples analyzed in COSMIC (May 2020). A study of Thai TNBC patients has reported KMT2C to be among the most frequently mutated genes in this population, found in 57.76% (*n* = 116) of all cases [68]. In a study by Serio et al., KMT2C was among the 15 most altered genes and was more frequently altered in elderly TNBC patients (26%, *n* = 22) than in young patients [69]. Li et al. investigated tissue samples of 17 Chinese TNBC patients with special morphologic patterns (STs). Here, KMT2C was altered in 18.8% and thus ranged among the top five altered genes as well [70]. In our cohort, KMT2C was altered in cfDNA of all 15 plasma and the matching urine samples, which appeared to be more frequent compared to the tissue samples investigated in the studies mentioned above. However, the cohort in our pilot study was very small and thus not representative.

Our study cohort was very small, and a larger trial is required to further investigate the usage of urinary ctDNA and to evaluate how far personalized treatment based on mutational characteristics might be implemented during a clinical routine. However, to our knowledge, this is the first study comparing plasma and urinary cfDNA from TNBC patients using such a comprehensive gene panel for targeted sequencing. Until now, only a few studies were published investigating cfDNA from plasma and matching urine samples to gain a comprehensive insight into tumor heterogeneity, mostly focusing on alterations in single genes though. As mentioned earlier, Zuo et al. compared cfDNA recovered from both plasma and urine specimens of 250 patients with early breast cancer. A 98% concordance of plasma cfDNA and tumor tissue samples was observed, and no false positive results were recorded. A similar level of agreement (97%) was reported for urinary DNA. There was also a strong agreement among the plasma and urine testing results, with 240 samples showing identical findings [41]. However, it has to be considered that their mutational analysis was based on one common PIK3CA variant only. Liu et al. also focused on PIK3CA and investigated cfDNA derived from the plasma vs. urine of 200 patients with early breast cancer. Molecular analysis showed that patients were successfully identified with mutant PIK3CA using urinary DNA. A strong correlation was affirmed from urinary and plasma DNA at baseline with the correlation coefficient *r* = 0.859 [40].

Several studies have already shown the good potential of cfDNA for monitoring treatment response or early detection of metastasis and recurrence [54,71,72]. However, application of ctDNA in the early diagnosis of breast cancer remains highly challenging due to the contamination of background DNA and relatively low levels of ctDNA [73]. Often, the concentration of ctDNA is only about 1 to 10  ng/mL in asymptomatic individuals. Therefore, in order to achieve 95% sensitivity, it was shown that an around 150 to 300 mL blood sample per test would be needed for breast cancer screening [74,75]. Although highly sensitive and specific methods have been developed to detect ctDNA, ultra-sensitive technologies that can detect the minimum amount of ctDNA are urgently needed to make this approach useful for early diagnosis [76].

Urine-based tests are non-invasive and therefore very patient-friendly. They are the most convenient for the patients as they could collect their samples at home and ship them to clinical laboratories. Novel preservative reagents were developed to stabilize cell-free DNA (cfDNA) in urine samples during storage and transportation for up to 7 days at various temperatures up to 37 °C. There would be no need for frequent visits in the outpatient unit, which could save time and cost, for both the patient and the clinic. Neither special equipment nor trained medical staff would be required for sampling. Further studies with a larger patient cohort, using proper sequencing parameters and adequate bioinformatics algorithms, are required to find associations between somatic variants in both body fluids and to investigate whether urine might be an additional or even an alternative cfDNA source in the future.

## 5. Conclusions

Taken together, our results indicated that both plasma-derived and urinary cfDNA from TNBC patients could be analyzed in a sufficient manner using a targeted sequencing approach. Both body fluids appear to be valuable sources bearing complementary information regarding the genetic tumor profile, which might be relevant for disease monitoring and individual treatment decisions. The results of the present study suggest using urinary cfDNA as a complementary analysis to plasma cfDNA in order to gain even deeper insights into tumoral heterogeneity in the future.

## Figures and Tables

**Figure 1 cancers-14-04101-f001:**
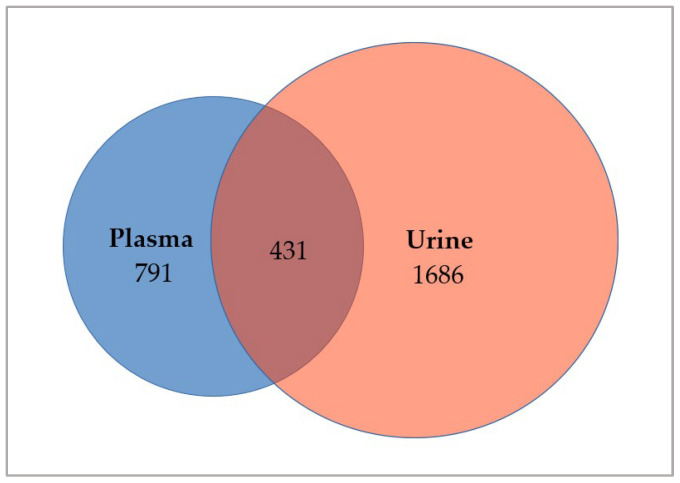
Venn diagram showing the number of somatic variants that could be detected in the ctDNA of both body fluids. Comparing all ctDNA variants of matching plasma and urine samples, 791 were exclusively found in plasma ctDNA, while 1686 variants were found in urinary ctDNA only. Interestingly, a total number of 431 were found in the ctDNA of both body fluids.

**Figure 2 cancers-14-04101-f002:**
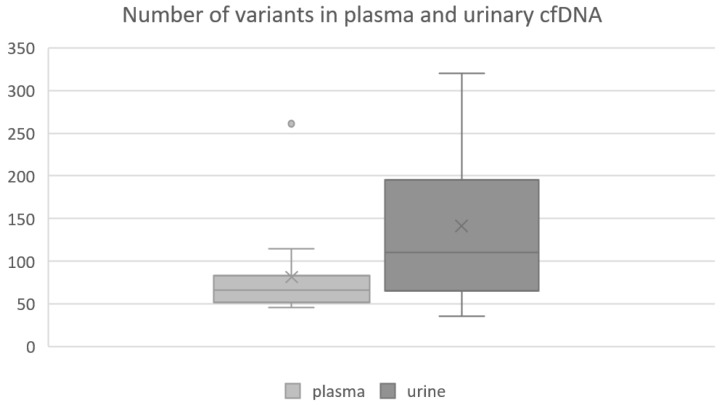
Boxplot showing the number of variants found in plasma and urinary ctDNA. Per plasma sample, a median number of 66 variants were found in ctDNA, ranging from 46 to 261 variants. In contrast, a median number of 110 variants were obtained per urinary ctDNA sample, ranging from 36 to 320 variants. Medians and interquartile ranges are displayed. The mean value is marked with an x.

**Figure 3 cancers-14-04101-f003:**
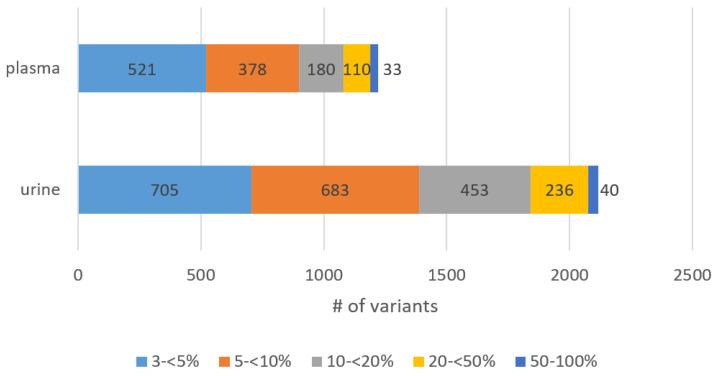
Stacked bar chart showing the VAF distribution in plasma- vs. urine-derived ctDNA. All variants were assigned to five different groups: VAF 3 ≤ 5%, 5 ≤ 10%, 10 ≤ 20%, 20 ≤ 50% and 50–100%. Even though we did not have matched germline samples for our cohort, 88% (*n* = 1079) of plasma-derived and 87% (*n* = 1841) of urinary-derived variants showed a VAF below 20%, indicating that the variants are rather of somatic than germline origin. In total, 98% of all variants (*n* = 3266) displayed a VAF below 50%, which additionally supports this thesis, as germline variants would most likely display a VAF of 50% or 100%.

**Figure 4 cancers-14-04101-f004:**
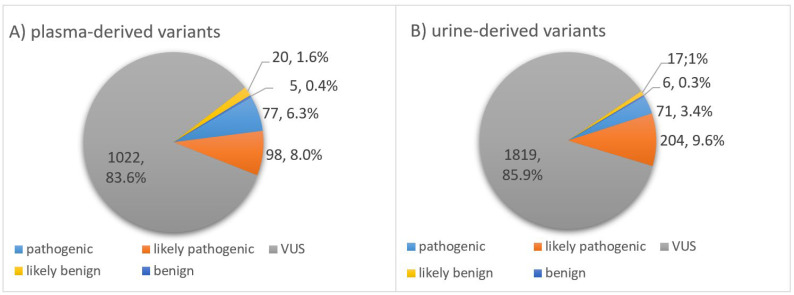
Impact. Classification of somatic variants found in ctDNA of (**A**) plasma and (**B**) matching urine samples of 15 TNBC patients after application of a personalized filter cascade. Using QCI Interpret Translational (QIAGEN), variants were classified into five subgroups according to their impact: pathogenic, likely pathogenic, variant of uncertain significance (VUS), likely benign or benign.

**Figure 5 cancers-14-04101-f005:**
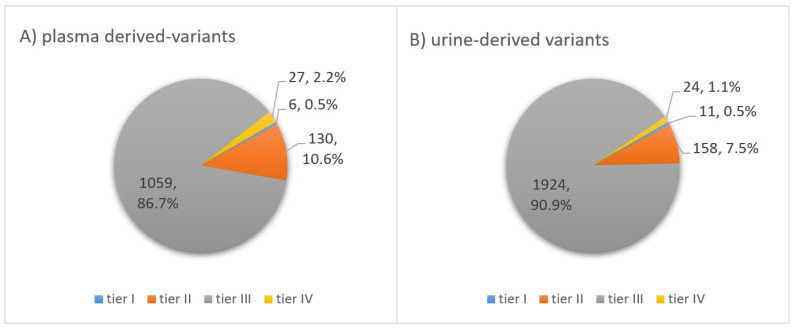
Clinical Significance. Classification of somatic variants found in ctDNA of (**A**) plasma and (**B**) matching urine samples according to their clinical significance. Using QCI Interpret Translational (QIAGEN), variants were classified into four tiers: Variants in tier I are of strong clinical significance (level A and B evidence), variants in tier II are of potential clinical significance (level C and D evidence), variants in tier III are of unknown clinical significance and variants in tier IV are characterized to be benign or likely benign.

**Figure 6 cancers-14-04101-f006:**
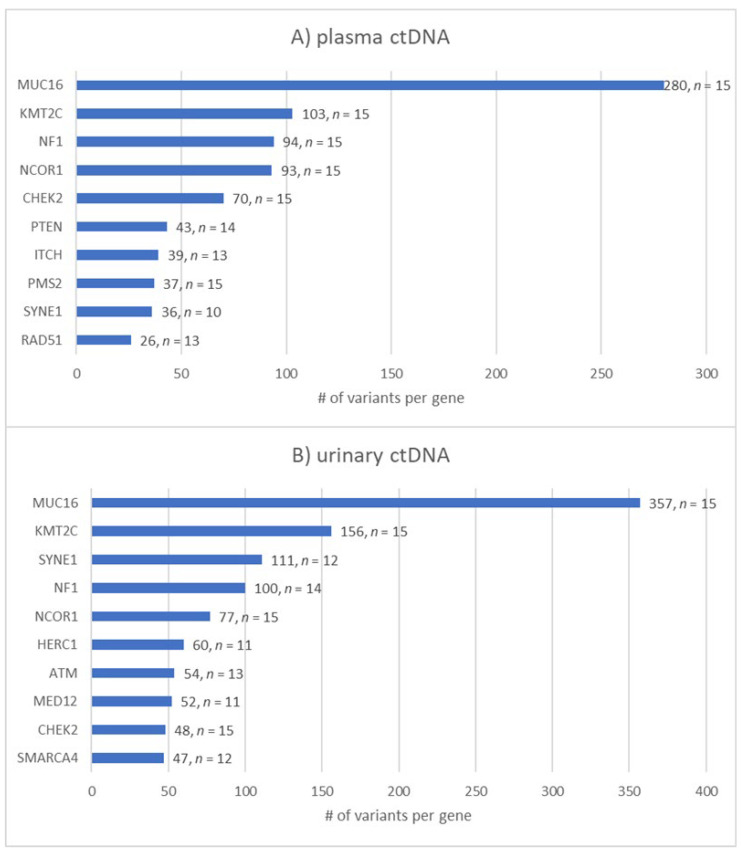
The top 10 most frequently mutated breast-cancer-associated genes in (**A**) plasma and (**B**) urinary ctDNA. Bar chart showing the number of variants found for the top 10 genes and the number of samples (*n* of 15) in which the gene was altered. MUC16 and KMT2C are by far the most altered genes in both ctDNA sources followed by NF1 in plasma and SYNE1 in urinary ctDNA.

**Figure 7 cancers-14-04101-f007:**
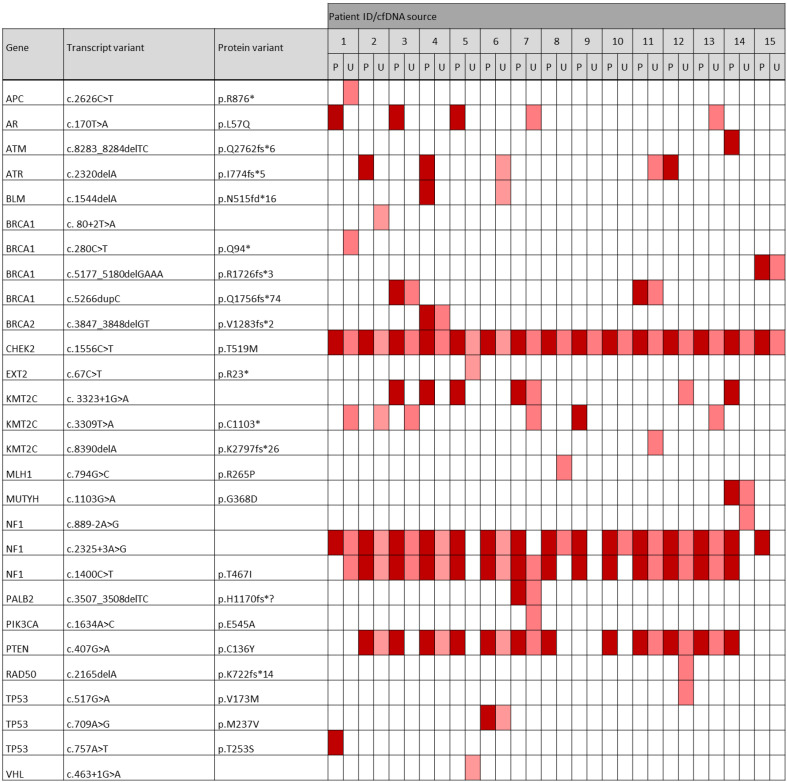
Heatmap of all pathogenic variants found in plasma ctDNA (*p*; dark red) and ctDNA of the matching urine samples (U; light red). In both body fluids, a median of 5 pathogenic variants were found, ranging from 3–8 variants in plasma ctDNA and from 1–8 in urinary ctDNA. Notably, when comparing pathogenic variants found in plasma and the matching urine sample per patient, 63% of pathogenic plasma variants could be detected in the matched urine samples and vice versa, and the recovery rate of pathogenic urine-derived variants in plasma cfDNA was 66%.

**Table 1 cancers-14-04101-t001:** Patient characteristics at baseline.

Characteristics	*n* = 15	%
Age at diagnosis		
Median (years)	48	
Range (years)	26–69	
KI-67 proliferation rate		
Median (%)	75	
Range (%)	10–90	
Histology		
Ductal	14	93
Medullary	1	7
Grade		
G2	4	27
G3	11	73
Tumor size		
T1	7	47
T2	7	47
T3	0	0
T4	1	7
Nodal status		
cN0	12	80
≥pN1	3	20
BRCA1/2 germline mutation		
yes	9	60
no	5	33
unknown	1	7
Neoadjuvant systemic therapy		
yes	13	87
*EC * + Paclitacel/Carboplatin*	*9*	*60*
*AC * + Paclitacel/Carboplatin*	*2*	*13*
*Docetaxel + Carboplatin*	*2*	*13*
no	2	13
Pathologic complete response after NACT (*n* = 13)		
yes	12	92
no	1	8

* EC: epirubicin + cyclophosphamide; AC: doxorubicin + cyclophosphamide; NACT: neoadjuvant chemotherapy.

**Table 2 cancers-14-04101-t002:** cfDNA concentrations from matched plasma and urine samples of 15 TNBC patients prior to surgical tumor resection in ng/mL. The median cfDNA concentration from 4 mL plasma eluted in 20 µL ultraclean water was 172 ng/mL, ranging from 37–442 ng/mL. The median cfDNA concentration from 10 mL urine eluted in 20 µL was 196 ng/mL, ranging from 0–1820 ng/mL. Initially, in two cases, cfDNA was not detectable.

Patient ID #	cfDNA Concentration (ng/mL)	
	Plasma	Urine
1	37	206
2	218	160
3	80	180
4	356	412
5	252	0
6	172	1820
7	244	196
8	420	0
9	69	650
10	212	192
11	118	1730
12	442	1410
13	112	180
14	158	110
15	80	254
median	172	196
range	37–442	0–1820

**Table 3 cancers-14-04101-t003:** Number of pathogenic/likely pathogenic variants per gene found in plasma and urinary ctDNA.

Gene	Variants in Plasma (*n*)	Variants in Urine (*n*)
NF1	49	54
KMT2C	25	32
CHEK2	25	19
PTEN	25	13

## Data Availability

The original raw sequencing data that support the findings of this study have been uploaded as fastq files and are available at the Sequence Read Archive (SRA) at GeneBank under BioProject ID PRJNA844219.

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
