# Peer review of "Targeted Sequencing of Plasma-Derived vs. Urinary cfDNA from Patients with Triple-Negative Breast Cancer"

_cancers, 2022, doi:10.3390/cancers14174101_

Round 1

Reviewer 1 Report

Herzog et al. analyzed plasma-derived and matching urinary cfDNA samples obtained from 15 presurgical triple negative breast cancer patients. The results showed the difference of somatic breast-cancer related variants in plasma-derived and urinary cfDNA. And their results indicated that body fluids may be valuable sources bearing complementary information regarding the genetic tumor profile. The topic is interesting, and there are some comments to the authors.

Comments:

1.       What are the main factors affecting cfDNA concentrations? The author needs to discuss them in the discussion section.

2.       The results showed that a total number of 431 was found in the ctDNA of both body fluids. Which tissues do these ctDNA mainly originate from? What evidence is there to support it. Which signaling pathways are these altered genes mainly enriched in?

3.       MUC16 and KMT2C are the most altered genes in both ctDNA sources. Whether these results are consistent with the genetic changes in TNBC patient tissues needs to be discussed.

4.       Utility of plasma and urinary ctDNA to monitor the breast cancer is still a huge challenge. Which of blood ctDNA or urine ctDNA is more suitable for early diagnosis of TNBC? Please discuss in depth in the discussion section.

Reviewer 2 Report

Henrike Herzog et al., investigated Targeted sequencing of plasma-derived vs. urinary cfDNA 2

from patients with triple negative breast cancer   The study is well-performed and the discussion is well-organized and comprehensive. Nevertheless, there are certain factors that must be fixed before publication:

1- Aim of the work should be stated clearly in introduction

            2- The purity of starting materials which used in the investigation should be added

3-How DNA is kept? Targeted sequencing of plasma-derived vs. urinary cfDNA 2

from patients with triple negative breast cancer

4- Comparison with related studies are is  recommended

        8-Finally, Please review English grammar before publication.

Round 2

Reviewer 1 Report

The authors have been replied the reviewer's comments point to point. And the revised manuscript was improved.